# New Insights into the Role of Ferroptosis in Cardiovascular Diseases

**DOI:** 10.3390/cells12060867

**Published:** 2023-03-10

**Authors:** Anna Maria Fratta Pasini, Chiara Stranieri, Fabiana Busti, Edoardo Giuseppe Di Leo, Domenico Girelli, Luciano Cominacini

**Affiliations:** Department of Medicine, Section of Internal Medicine D, University of Verona, 37134 Verona, Italy

**Keywords:** antioxidants, cardiovascular diseases, ferroptosis, iron overload, oxidative stress

## Abstract

Cardiovascular diseases (CVDs) are the principal cause of disease burden and death worldwide. Ferroptosis is a new form of regulated cell death mainly characterized by altered iron metabolism, increased polyunsaturated fatty acid peroxidation by reactive oxygen species, depletion of glutathione and inactivation of glutathione peroxidase 4. Recently, a series of studies have indicated that ferroptosis is involved in the death of cardiac and vascular cells and has a key impact on the mechanisms leading to CVDs such as ischemic heart disease, ischemia/reperfusion injury, cardiomyopathies, and heart failure. In this article, we reviewed the molecular mechanism of ferroptosis and the current understanding of the pathophysiological role of ferroptosis in ischemic heart disease and in some cardiomyopathies. Moreover, the comprehension of the machinery governing ferroptosis in vascular cells and cardiomyocytes may provide new insights into preventive and therapeutic strategies in CVDs.

## 1. Introduction

Cardiovascular diseases (CVDs) still remain the leading cause of disease burden and premature death worldwide [1,2]. Recent data indicated that in 2019 the prevalence of CVDs was up to 523 million, and the number of deaths correlated to CVDs reached 18.6 million principally caused by ischemic heart disease (49%) and stroke (17.7%) [3]. In the last decades, different forms of regulated and unregulated cardiac and vascular cell death have been implicated in the pathogenesis of multiple forms of heart disease ranging from myocardial infarction (MI), heart failure of diverse etiologies, myocarditis, and congenital heart disease [4,5,6]. Ferroptosis is an iron-dependent form of non-apoptotic cell death that involves the accumulation of lipid hydroperoxides resulting in oxidative damage of cell membranes [7]. From a morphological and mechanistic point of view, ferroptosis is different from the other forms of regulated cell death [8]. Not long ago, a series of studies suggested that ferroptosis plays a role in the death of cardiac and vascular cells induced by different stresses [9,10,11]. Therefore, ferroptosis may have an impact on the mechanisms leading to CVDs and become a suitable target for their prevention [12,13]. In this article, we review the emerging role of ferroptosis in heart disease. In particular, we summarize the molecular mechanism of ferroptosis and the current understanding of the pathophysiological role of ferroptosis in ischemic heart disease and in cardiomyopathies.

## 2. Molecular Mechanisms of Ferroptosis

Cell death correlated to ferroptosis can be traced back to the 1950s when Eagle [14] reported that the shortage of cystine hampers cell growth and leads to cell death. However, only 50 years later, in 2003, Dolma et al. [15] reported that erastin, a compound possessing lethal activity against cells expressing small GTPase RAS, could selectively begin a process of cell death, with no apoptotic characteristics. Furthermore, in 2008, Yang and Stockwell [16] demonstrated that the inhibition of glutathione peroxidase (GPX) 4 with a substance called RAS-selective lethal 3 (RSL3) induced nonapoptotic cell death in tumorigenic cells expressing RAS. Interestingly, it was also shown that this sort of cell death could be counteracted by genetic blocking of cellular iron uptake or iron chelation [16,17]. In the end, in 2012, Dixon et al. [7] found that erastin caused a huge, iron-dependent generation of reactive oxygen species (ROS) leading to a new form of non-apoptotic cell death, which was called ferroptosis. In particular, erastin was shown to inhibit cystine uptake by the cystine/glutamate antiporter (System Xc-) with the depletion of glutathione (GSH) [7]. Since then, mounting studies have revealed that altered iron metabolism, depletion of GSH, inactivation of GPX4, and upregulation of polyunsaturated fatty acid (PUFA) peroxidation by ROS are crucial to the beginning and development of ferroptosis [7,18]. Furthermore, in 2017, it was reported that acyl-CoA synthetase long-chain family member 4 (ACSL4) is a critical contributor to ferroptosis, since it is required for the assembly of PUFAs [19]. Lately, a new ferroptosis halting pathway has been discovered with the recognition that ferroptosis suppressor protein 1 (FSP1), a coenzyme Q10 (CoQ10) oxidoreductase, can suppress ferroptosis in a GSH-independent way [20,21]. A scheme of the molecular mechanism of ferroptosis is shown in Figure 1.

### 2.1. Oxidation Mechanisms

PUFAs are constituents of the cell membrane and control a number of biological activities, such as inflammation, immunity, synaptic plasticity, and cellular growth [23]. PUFAs are susceptible to oxidation as a result of the weak C-H bond at the bis-allylic positions [24] which are the main target of ROS [25]. Interestingly, exogenous delivery of monounsaturated fatty acid oleic acid can suppress ferroptosis induced by erastin by competing with PUFAs for incorporation into phospholipids suggesting that monounsaturated fatty acids are not a target of ROS during ferroptosis [26]. Recently, several lipidomic reports showed that phosphatidylethanolamines (PEs) incorporating arachidonic acid (AA) or its elongation derivative, adrenic acid, are key phospholipids (PLs) that undergo oxidation and trigger ferroptosis [27,28]. Coenzyme-A-by-products take part in the synthesis of these PUFAs and their positioning into PLs to start ferroptosis [8]. In this context, two enzymes, ACSL4 and lysophosphatidylcholine acyltransferase 3 (LPCAT3), have been shown to take part in the generation and assembly of PUFA-PEs in the cell membrane [29]. ACSL4 acylates AA and then LPCAT3 inserts the acylated AA into membrane PLs, a fact that augments the probability that sensitive PUFAs can be oxidized and eventually triggers ferroptosis [30]. In summary, taking AA as an example, ACSL4 causes the ligation of CoA into AA to form a CoA-AA intermediate, which is then esterified into PE by LPCAT3 to generate PE-AA. Several hypotheses have been suggested about the cell death mechanisms induced by ferroptosis. It is known that PUFAs are essential constituents of the cell membranes, and the chemical and geometric configuration of the lipid bilayer could be modified by heavy lipid peroxidation [31]. In addition, peroxide cumulation could result in the generation of membrane pores and disrupt the barrier function with the alteration of membrane permeabilization [31].

Up to now, the mechanisms initiating lipid peroxidation are not entirely known, but can potentially occur through non-enzymatic and/or enzymatic processes [23]. Current reports suggest that different metabolic events in mitochondria have significant roles in triggering ferroptosis. First of all, it is known that mitochondrion is an important source of cellular ROS [32]. In particular, superoxide (O2^•−^) is produced by the electron leakage from electron transport chain (ETC) complexes I and III. Thereafter, O2^•−^ is transformed into hydrogen peroxide (H_2_O_2_) through the activity of mitochondrial superoxide dismutase (SOD) [32]. H_2_O_2_ interact with ferrous ion (Fe^2+^) to give rise via the Fenton reaction to hydroxyl radicals (^•^OH), which subsequently remove the PUFA bis-allylic hydrogen to produce PUFA radicals (PUFA^•^). These unstable radicals promptly react with oxygen to produce PUFA peroxyl radicals, which eventually generate PUFA hydroperoxides [31].

As for enzymatic processes involved in ferroptosis, lipoxygenases (LOXs) are a family of iron-containing enzymes that catalyze the deoxygenation of PUFAs. There are six LOX isoforms in human beings: 15-LOX-1, 15-LOX-2, 12-LOX-1, 12-LOX-2, E3-LOX, and 5-LOX, of which 12 and 15-LOX are broadly allocated in a variety of tissues [33]. The typical substrate of LOX are PUFAs, and the sn2-15-hydroperoxy-eicasotetraenoyl-phosphatidylethanolamines (sn2-15-HpETE-PE) generated by 15-LOX is regarded as a marker of ferroptosis [34]. Nevertheless, as noted below, the central role of LOX in triggering ferroptosis is still open to question. Since the preferential substrate of LOX are free PUFAs, LOX must first split PUFAs from PLs by means of a phospholipase [35]. However, this is incompatible with the established mechanisms of ferroptosis since the lipid peroxidation takes place on esterified PUFA-PLs instead of free PUFAs, and the implication of LPCAT3 in ferroptosis is in agreement with this view [29,36]. Furthermore, there are some reports showing that deletion of 12- or 15-LOX cannot retrieve the fatal lethality in GPX4 knockout mice [37]. In addition, it has been shown that the deletion of 12- or 15-LOX in adult mice carrying whole-body GPX4 deletion cannot inhibit kidney cell death [38]. Finally, some cell lines not expressing any of the major LOX enzymes resulted in being sensitive to ferroptosis [39]. Uncertain and not completely defined in triggering ferroptosis is the role of other oxygenases such as nicotinamide adenine dinucleotide phosphate oxidase (NADPH) and of cytochrome P450 oxidoreductase [40,41,42].

### 2.2. Antioxidant Mechanisms

#### 2.2.1. GPx4/GSH Axis

GPX4 is a member of the selenoprotein family, whose active sites reside in the amino acid selenocysteine [43]. GPX4 turns GSH into oxidized glutathione (GSSG) and transforms the cytotoxic LOOH to the corresponding alcohols [43]. GPX4 is a protein fulfiling several functions with the ability to reduce lipids peroxides either in uncoupled form or when combined with lipids such as PLs [44]. The catalytic reaction of GPX4 goes rapidly along with different states, commuting between oxidized and reduced states. In particular, selenolate in GPX4 is first oxidized to selenic acid (se-OH) by a peroxide substrate. Then, GSH reduces Se-OH to a reactive intermediate which is then reduced by a second molecule of GSH to form GSSG with the regeneration of the enzyme [45]. GSH can be restored by reducing GSSG through glutathione reductase and nicotinamide adenine dinucleotide phosphate oxidase (NOX) as the electron donor [46]. GSH requires cysteine for its synthesis from glutamate [47]; cellular importing of cystine is associated with the export of glutamate via system Xc- [7,47]. System Xc- is a cystine/glutamate antiporter that makes possible the interchange of cystine and glutamate across the plasma membrane [7]. The system includes the solute carrier family 7A member 11 subunit (SLC7A11) and the solute carrier family 3A member 2 subunit (SLC3A2) 119 [48], (Figure 1). Interestingly, while the expression of SLC7A11 is directly controlled by the transcription factor erythroid 2-related factor 2 (Nrf2) in situations of cellular stress [49], the tumor suppressor protein p53 inhibits SLC7A11 transcription, so favoring ferroptosis [50]. In addition, it has been demonstrated that the transcription factor ATF3 favors ferroptosis by inhibiting the expression of SLC7A11 in a p53-independent mode [51]. After cystine is delivered into the cell, it is reduced to cysteine by GSH or thioredoxin reductase 1 [52]. By inhibiting this system by the synthetic small molecule erastin or other compounds, the importation of cystine is suppressed [7,52]. The consequent deficiency of cysteine hinders GSH synthesis, and as a result, gives rise to increased glutaminolysis [18]. Excessive glutaminolysis activates mitochondrial tricarboxylic acid cycle activity and greatly boosts mitochondrial respiration, determining hyperpolarization and increased generation of ROS, which eventually stimulates lipid peroxidation [18,53]. Depletion of GSH and inactivation of GPX4, at last culminate in ungovernable lipid peroxidation and cell death through ferroptosis [50,51].

#### 2.2.2. GPX4-Independent Pathways

In addition to the classical GPX4/GSH-dependent anti-ferroptotic pathway, recent reports have discovered new regulatory routes distinct from GPX4/GSH axis, including the FSP1–CoQ10 pathway [20,21,54], the endosomal sorting complex required for transport-III (ESCRT-III)-mediated membrane repair pathway [55], and the guanosine triphosphate cyclohydrolase 1-tetrahydrobiopterin (GCH1-BH4) pathway [56]. FSP1 catalyzes the reduction in non-mitochondrial CoQ10 using NAD(P)H [20,21,54]. CoQ10, as a reversible redox carrier in the plasma membrane and Golgi apparatus membrane electron transport [57], is a crucial endogenous lipid-soluble antioxidant capable of scavenging lipid peroxyl radicals [58] and thus inhibiting ferroptosis. CoQ10, as the principal target of the FSP1 pathway, is diffused to a large degree amidst the membranes of mammalian cells [59].

ESCRT-III is a member of the ESCRT family complex, which is comprised of five subcomplexes and takes part in membrane remodeling. Dai et al. [55] found that standard inducers of ferroptosis-like erastin and RSL3 expand the ESCRT-III subunits gathering, while knocking down components of ESCRT-III machinery was shown to intensify ferroptosis.

GCH1 is regarded as the rate-limiting enzyme for the antioxidant BH4 generation. Evidence indicates that BH4 is a strong radical-trapping antioxidant that saves cells from ferroptosis by lessening lipid peroxidation, and it is restored by dihydrofolate reductase [60]. Recently, Kraft et al. [56], by performing a genome-wide activation screen, reported that GCH1 is the most important gene implicated in ferroptosis prevention, which is independent of GPX4/GSH axis activity.

### 2.3. Iron Metabolism and Ferroptosis

Although iron is required in many biological processes and crucial for cell viability, either iron overload or deficiency can lead to pathological situations. Ferroptosis is a kind of regulated cell death related to a surplus of iron that can induce lipid peroxidation and cell death [61]. Iron quantity and handling in the body are strictly regulated at both cellular and systemic levels [62]. Iron metabolism is a dynamic process entailing many steps, such as iron absorption, storage, utilization, and efflux. The nonheme insoluble iron Fe^3+^ in food needs to be reduced to Fe^2+^ to be absorbed; then, it is reoxidized in serum by multi-copper iron oxidase hephaestin or ceruloplasmin to ferric ion (Fe^3+^) to combine with transferrin (TF), the main extracellular iron-binding protein [63]. Fe^3+^ is carried into the cell by TF receptor 1 (TFR1), then transformed into Fe^2+^ in the endosome by six transmembrane epithelial antigens of the prostate 3 and liberated from the endosome by divalent metal transporter 1 (DMT1) [63]. Iron transported in cytoplasm is embedded into different iron-containing proteins or, if in excess, it is stowed in ferritin [30,63]. When required, Fe2^+^ is exported in circulation through the hepcidin-regulated protein ferroportin (FPN) [64], which is the only known mammalian protein able to transport intracellular iron outside the cell [65]. Exported Fe^2+^ is then oxidized to Fe^3+^ by ceruloplasmin bound to cell membrane and carried into circulation mainly through TF [66]. A number of studies reported that iron overload attributable to unbalance between iron import, accumulation, and export plays a central role in determining the susceptibility of cells to ferroptosis [7,18]. In this scenario, upregulation of the TFR1 has been shown to promote iron import, thus triggering ferroptosis [67]. Similarly, in FPN-deficient mice increased cellular iron favor ferroptosis [68]. Cellular ferritin can be broken down through an autophagic process, known as ferritinophagy, activated by nuclear receptor coactivator 4 (NCOA4) [69] (Figure 1). This process causes the degradation of lysosomal ferritin, with successive liberation of iron [70] and its export to the cytosol through lysosomal natural resistance-associated macrophage protein 2 [71]. Enhanced ferritin degradation through ferritinophagy could increase the level of the labile iron pool (LIP) and enhance ferroptosis [69]. In general, cellular iron balance is finely tuned by a post-transcriptional machinery through the iron regulatory proteins (IRP) 1 and 2, which interact with iron-responsive elements (IRE) on mRNA of several proteins involved in cellular iron uptake, storage, and release [72]. When iron concentration is inadequate, the mRNA of TFR and DMT-1 is stabilized by IRP to upgrade iron influx [73,74]. At the same time, IRP hampers mRNA of FPN1 and ferritin to prevent iron efflux and storage [73,74]. This course of actions elicits stability of LIP, that is a crucial crossroad of cellular iron metabolism. LIP in fact is in a transitional position between extracellular iron and cellular iron coupled with proteins [75]. When LIP concentration exceeds its homeostatic limits, the Fenton reaction can trigger lipid peroxidation which eventually may lead to ferroptosis [27,75,76].

It is now well-recognized that the maintenance of iron homeostasis in mitochondria also plays a key part in counteracting ferroptosis [77]. Mitoferrin 1 and mitoferrin 2 are essential importers of mitochondrial iron required for the synthesis of heme and Fe-S complex [78]. Mitoferrin 2 abrogation lowers cell death caused by erastin, while its overexpression rises ferroptosis [79]. In addition, mitochondrial HO-1 causes heme breakdown providing Fe^2+^, giving rise to mitochondrial iron overload that can lead to ferroptosis [10,80,81]. Similarly to ferritin in cytosol, mitochondrial ferritin also protects against ferroptosis: cells overexpressing mitochondrial ferritin have been shown to be unaffected by ferroptosis induced by erastin [81]. In addition, it has been demonstrated that the mitochondrial Fe-S-binding protein mitoNEET is involved in mitochondrial iron transporting [82,83]. In particular, mitoNEET has been reported to inhibit mitochondrial iron transport into the matrix [84]. Since iron is a rate-limiting component for electron transport, mitoNEET activity is associated with reduced mitochondrial membrane potential and lower ROS generation [84]. As a matter of fact, enhanced expression of mitoNEET counteracted ferroptosis induced by erastin in human hepatocellular carcinoma cells [85]. Furthermore, Mao et al. [86] have recently furnished firm proves that dihydroorotate dehydrogenase-CoQ10 together with mitochondrial GPX4 serve as a mitochondrial defense system. This framework was shown to inhibit mitochondrial lipid peroxidation and limit ferroptosis. Dihydroorotate dehydrogenase, in fact, is a mitochondrial enzyme involved in pyrimidine production, which combines the oxidation of dihydroorotate to orotate, to the reduction of CoQ10 to CoQ10H2 which serves as a radical-trapping antioxidant [87,88].

## 3. Ferroptosis and CVDs

### 3.1. Atherosclerosis, MI and Ischemia-Reperfusion (I-R) Injury

Atherosclerosis underlies the pathophysiology of ischemic heart disease which remains the major cause of death and disability globally [89]. Atherosclerosis is considered a chronic inflammatory condition initially characterized by endothelial cell and smooth muscle cell dysfunction and the appearance of macrophages, which account for atherosclerotic plaque development. The accumulation of oxidized lipids in the arterial wall is also considered a key element in atherosclerosis [90,91]. Moreover, significantly increased levels of iron in atherosclerotic lesions have been observed both in human and animal models [92]. Evidence indicates that excessive iron, besides triggering ferroptosis, can worsen oxidative stress in cells and induce the expression of some adhesion molecules, such as intercellular cell adhesion molecule-1, vascular cell adhesion molecule-1, and monocyte chemoattractant protein-1, thus aggravating endothelial damage and promoting lipid deposition [93,94,95,96]. Furthermore, the iron-induced free radical generation, together with the increased expression of ferroptosis-associated LOXs [97] can promote low-density-lipoprotein (LDL) oxidation [98,99,100]. Oxidized LDL can be taken by macrophages to form foam cells which in turn upregulate some proteolytic enzymes degrading the extracellular matrix structure and leading to atherosclerotic plaque disruption and eventually to clinical severe cardiovascular events. Interestingly, inhibition of 12/15 LOXs significantly reduced oxidized LDL subendothelial deposition and weakened atherosclerosis progression [97]. Since uncontrolled lipid peroxidation and iron overload are the main hallmarks of ferroptosis, it is plausible that ferroptosis might cooperate in the initial and progressive steps of the atherosclerotic process. In this context, Guo et al. [101] aimed to evaluate the effect of GPX4 overexpression on aorta atherosclerotic lesions in high-fat diet (HFD)-induced atherosclerosis in apolipoprotein (Apo) E-deficient mice. The results provide evidence that GPX4 inhibits the development of atherosclerosis by decreasing lipid peroxidation and decreasing the sensitivity of vascular cells to oxidized lipids. Bai et al. [102] administered the ferroptosis inhibitor ferrostatin-1 (Fer-1) [7,103] to HFD-ApoE deficient mice. The results indicated that the decreased expression of SLC7A11 and GPX4, two critical key factors in counteracting ferroptosis [44,45,46,49], was in part mitigated by Fer-1. These data suggest that Fer-1 could alleviate atherosclerotic lesions and partially inhibit iron accumulation and lipid peroxidation. In the context of human coronary atherosclerosis, Zhou et al. recently investigated the expression of ferroptosis-related proteins prostaglandin-endoperoxide synthase 2 (PTGS2), ACSL4, and GPX4 in different stages of coronary atherosclerosis. They found that PTGS2 and ACSL4 were upregulated, while GPX4 was downregulated in the advanced atherosclerotic plaques of human coronary arteries, indicating that ferroptosis may also contribute to plaque instability in humans [104]. Furthermore, overexpression of TFR1 was found to be significantly expressed in foam cells, a fact that contributes to iron accumulation and ferritin synthesis, two events that favor atherosclerotic plaque rupture [105]. Finally, Habib et al. reported that hepcidin/FPN axis and, in particular, a reduction in FPN-related iron export, controls TLR-dependent macrophage inflammatory response which plays a crucial role in atherosclerosis progression [106].

It is well recognized that sustained ischemia in MI often results from thrombotic occlusion of a coronary artery following rupture/erosion of an atherosclerotic plaque [107]. MI is accompanied by numerous structural and functional consequences, the most irreversible of which is myocardial cell death [108]. Interestingly, clinical findings have proposed that myocardial iron is a risk factor for left ventricular remodeling in patients after MI [109]. Although myocardial cell death is the initiating and central cardiac event in MI, its underlying mechanism remains to be fully explained. Currently, some experimental models have shed light on the possible involvement of ferroptosis during MI. A recent study using proteomic analysis found a down-regulation of GPX4 with subsequent ferroptosis in the early and middle stages of the MI mice model; moreover, the deletion of GPX4 caused the accumulation of lipid peroxides, leading to cardiomyoblast cell death by ferroptosis [110]. Furthermore, HO-1 activity was shown to be enhanced via the Nrf2/HO-1 pathway in the early and middle stages of MI, leading to the iron excess that contributed to ferroptosis in cardiac cells [13,111]. Further experimental studies have explored the possible effect of specifically targeting excess iron in MI. In this context, Baba et al. [112] reported a protective effect of the mechanistic target of rapamycin and of Fer-1 against iron overload-related ferroptosis by lowering ROS generation in cardiomyocytes. Evidence indicates that mesenchymal stem cell (MSC)-derived exosomes mediate tissue regeneration in ischemic heart injury [113] and that miR-23a-3p, a kind of enriched miRNA present in MSC-derived exosomes, plays a key role in angiogenesis and vasculature development [114]. In addition, MSC exosomes derived from human umbilical cord blood were demonstrated to suppress ferroptosis via miR-23a-3p/DMT1 axis and to mediate myocardial repair in MI mice [115]. These studies indicate that ferroptosis is involved not only in the occurrence and development of atherosclerosis but also in the mechanisms leading to plaque rupture and adverse clinical events, such as MI. The opportunity to target ferroptosis could open a new potential scenario for the prevention of ischemic heart disease.

It is well known that the objectives of MI therapy are to lower infarction size, to maintain the systolic function of the left ventricular, and to hamper the onset of heart failure [116,117]. Restoration of the blood perfusion of ischemic myocardium as soon as possible is the current major strategy for MI treatment [118,119]. However, like ischemia, reperfusion itself can paradoxically cause myocardial damage, known as ischemia-reperfusion (I/R) injury [119]. I/R injury is a quite usual, clinical complication that can take place in just about any organ, including the heart, liver, kidneys, and brain [77]. Although several studies have shown that ferroptosis is a key cause of renal and hepatic I/R injury [36,120,121], ferroptosis in myocardial I/R injury has been the most extensively studied [122,123,124,125,126,127]. An interesting study showed that ferroptosis primarily occurs during the period of myocardial reperfusion rather than ischemia [123]. Indeed, in the course of ischemia, no considerable differences were found in ferroptosis markers such as GPX4, ACSL4, Fe^2^+, and malondialdehyde (MDA) in heart tissue. However, during ischemia, specific redox reactions of PUFA-phospholipids in cardiomyocytes are induced, which act as initiating signals to trigger intensive oxidative damage during the reperfusion phase [128]. More specifically, it has been reported that PUFA-PEs are oxidized by LOX-15, an enzyme that is induced by hypoxia [129]. Moreover, PUFA-PEs are also oxidized by the excess of ROS produced by mitochondria [129] and by the accumulation of local iron in injured heart tissue [130]. In addition, the level of NOX2, a ROS generator enzyme [40], has been shown to be increased during myocardial I/R injury [131]. Thereafter, only when the perfusion time is prolonged, a progressive elevation of ACSL4, Fe^2+^, and MDA, coupled with a decline of GPX4, was detected [123]. Furthermore, Ma et al. [132] reported that the expression of antioxidant sirtuin 1 (SIRT1) in the heart tissue was repressed after I/R damage, and SIRT1 overexpression was shown to lower the extent of I/R injury-related ferroptotic cell death via p53/SLC7A11 axis. Machado et al. [133] reported that during cardiac I/R injury, ferritin shortage activates the synthesis of many antiferroptotic proteins, including HO-1. In turn, HO-1 upregulation leads to SLC7A11 induction and hence to GSH generation which reduces ferroptosis and preserves heart function. Nevertheless, HO-1 upregulation may be a double-edged sword since it causes heme degradation and the release of Fe^2+^ which promotes ferroptosis [133]. Finally, it has to be noted that ubiquitination and some non-coding RNAs have also been shown to play a role in ferroptosis induced by I/R injury [134].

Growing experimental studies show that targeting ferroptosis may be a promising protective treatment for myocardial I/R injury [10,124,125,126,127]. Inhibition of glutaminolysis was first shown to reduce I/R-induced heart injury in an ex vivo heart model [9]. Further in vivo evidence demonstrated that either ferroptosis inhibition or iron chelation during both acute and chronic myocardial I/R injury can give protective effects. Resveratrol was recently shown to protect against myocardial I/R injury via reducing oxidative stress and attenuating ferroptosis both in vitro and in vivo [124]. Histochrome, a potent antioxidant with iron-chelating properties, was shown to prevent ferroptotic cell death in rat myocardial I/R injury by decreasing ROS levels and by upregulating the expression of Nrf2 and antioxidant genes [125]. Schematic mechanisms leading to ferroptosis in ischemic heart disease and I/R injury are shown in Figure 2.

### 3.2. Ferroptosis and Cardiomyopathies

Cardiomyopathy is a progressive heart disease with multifactorial pathogenesis and high mortality. The definition and classification of cardiomyopathy have evolved considerably in recent years. The American Heart Association proposed in 2006 a classification that divides cardiomyopathies into primary (genetic, mixed, or acquired) and secondary causes including ischemic, metabolic, infectious, toxic, auto-immunogenic, and neuromuscular [134]. According to a position statement from the European Society of Cardiology, cardiomyopathy is a myocardial disorder in which the heart muscle is structurally and functionally abnormal, in the absence of coronary artery disease, hypertension, valvular disease and congenital heart disease sufficient to cause the observed myocardial abnormality [135]. In this review we will focus on the following cardiomyopathies: diabetic cardiomyopathy [136], hypertrophic cardiomyopathy, dilated cardiomyopathy [137], secondary hypertrophic cardiomyopathy [74], post-transplant cardiomyopathy [138], iron overload cardiomyopathy [139], septic cardiomyopathy [140], doxorubicin-induced cardiomyopathy [141], and radiation-induced cardiomyopathy [142] in light of their high clinical relevance and the available data so far published. All of these cardiomyopathies are associated, by definition, with cardiomyocyte death. The injured tissue is progressively substituted by fibrotic tissue with no contractile function [143]. In the end, the depletion of cardiomyocytes can result in pathological ventricular remodeling and heart failure.

#### 3.2.1. Diabetic Cardiomyopathy

The number of people with diabetes is estimated to increase from 463 million in 2019 to 700 million by 2045 [144]. The first clinical report of cardiomyopathy associated with type 2 diabetes mellitus was described approximately 50 years ago [145] and still remains the main cause of heart failure in individuals with diabetes [146]. Diabetic cardiomyopathy is considered a disruption of heart structure and function in diabetic patients in the lack of ischemic heart disease, hypertension, valvular heart diseases, and other typical cardiovascular diseases. Diabetic cardiomyopathy is characterized in its early stages by diastolic relaxation abnormalities and later by clinical heart failure [136].

Evidence indicates that cell death plays a key role in diabetic cardiomyopathy pathogenesis [147]. Although apoptosis has been shown to be predominant in cardiomyocyte death in the early phases of diabetes [148,149,150,151], recent data show that the contribution of ferroptosis in the progression of the disease is critical [152,153]. Advanced glycation end-products (AGEs) are glycated proteins or lipids forged in the presence of hyperglycemia [150,151,154]. AGEs are enhanced in diabetic patients carrying diabetic cardiomyopathy, and AGEs are regarded as the predominant factors in initiating diabetic cardiomyopathy by their binding to the AGE receptor [155,156,157,158,159]. Previous studies on cultured cells have demonstrated that AGEs increased the levels of PTGS2 [160,161], a new marker of ferroptosis. A recent paper provided evidence that ferroptosis is crucial in the pathogenesis of diabetic cardiomyopathy in mice and in a new ex vivo diabetic cardiomyopathy model utilizing engineered cardiac tissues (ECTs) [153]. In particular, it has been shown that intracellular ferritin levels were temporarily high in ECTs treated with AGEs, but in the subsequent 24 h, ferritin declined [153], a circumstance that favors ferroptosis [7]. In addition, in both AGE-treated ETC and the hearts of diabetic rodents, LIP was found to be elevated [150], once again an event that promotes ferroptosis [7]. Iron overload produced by ferritin reduction and LIP elevation through the Fenton reaction can burst ROS generation [161] which in turn raises AGE formation, sustaining a vicious cycle [162]. Finally, Wang et al. [153] reported that SLC7A11 and GSH were significantly reduced in AGE-treated ECTs and hearts of mice with diabetic cardiomyopathy. Altogether, these pathological modifications might interfere with usual cell function and trigger lipid peroxidation, resulting in ferroptosis, and hence in diabetic cardiomyopathy development. Interestingly, Nrf2 activated by sulforaphane was shown to suppress cardiac cell ferroptosis in both AGE-treated ECTs and the hearts of diabetic cardiomyopathy mice by upregulating ferritin and SLC7A11 [150]. These protective effects on ferroptosis have been confirmed using other Nrf2 activators such as curcumin [163] and 6-gingerol [164] in animal models and in cardiomyocytes. Nevertheless, as emphasized above, it is known that Nrf2 increases HO-1 which degrades heme and makes available free iron that additionally favors lipid peroxidation [10]. Therefore, while ferroptosis is likely to play a critical role in the pathogenesis of diabetic cardiomyopathy, the detailed mechanisms leading to cardiomyocyte death and to ferroptosis need to be further explored. Schematic mechanisms leading to ferroptosis in diabetic cardiomyopathy are shown in Figure 2.

#### 3.2.2. Primary Hypertrophic Cardiomyopathy and Dilated Cardiomyopathy

Hypertrophic cardiomyopathy is the most frequent primary cardiomyopathy occurring in 1:500 individuals [137]. It is a familial cardiomyopathy related to myosin heavy chain and sarcomeric protein gene mutations [165,166]. The peculiar characteristics of hypertrophic cardiomyopathy are the presence of left ventricular hypertrophy [137] and its association with sudden cardiac death [167]. It is known that the L-type Ca^2+^ channel (LTCC) activity is augmented in hypertrophic cardiomyopathy myocardial cells [168]; this activation is associated with a larger intracellular Ca^2+^ which may be the cause of the arrhythmogenic status leading to sudden cardiac death [168,169]. As for the involvement of ferroptosis in hypertrophic cardiomyopathy, it has been demonstrated that LTCC activation not only enables extracellular Ca^2+^ entry, but also increases the intracellular accumulation of Fe^2+^ through the non-transferrin-bound iron (NTBI) uptake [170,171]. Accordingly, the concentration of intracellular iron is persistently high in hypertrophic cardiomyopathy, a fact that promotes excessive ROS generation and lipid peroxidation, exhaustion of antioxidants and finally ferroptosis. Furthermore, it has been previously established that hypertrophic cardiomyopathy is characterized by an altered glucose utilization and a prevalence of glycolysis with the production of pyruvate and lactate [172,173] that promote ferroptosis [174,175]. Fang et al. [176] recently reported that SLC7A11, a key regulator of GSH production [7,47] was downregulated in ferritin-deficient mice fed a high iron diet so favoring ferroptosis-related left ventricular hypertrophy. The downregulation of SLC7A11 was related to the activity of the p53 pathway, SLC7A11 being a newly discovered p53 target gene [50]. The same mechanism may be operating also in hypertrophic cardiomyopathy and favor ferroptosis.

Dilated cardiomyopathy is the second-most-frequent nonischemic cardiomyopathy with a frequency of 1:2500 [137]. Dilated cardiomyopathy patients are characterized by severe symptoms due to heart failure and sudden cardiac death and very often the only remedy is heart transplantation [137,177]. The mechanisms behind the disease can be genetic or due to secondary cardiomyopathies. The genetic form accounts for only 20–30% of dilated cardiomyopathy while the bulk of dilated cardiomyopathy is secondary to cardiomyopathies such as ischemic heart disease, diabetic cardiomyopathy, septic cardiomyopathy, etc. [178,179]. The mechanisms which contribute to ferroptosis in the genetic dilated cardiomyopathy are similar to those described in hypertrophic cardiomyopathy [180]. Schematic mechanisms leading to ferroptosis in hypertrophic and dilated cardiomyopathies are shown in Figure 3.

#### 3.2.3. Secondary Hypertrophic Cardiomyopathy

In the heart, pressure overload caused by pulmonary hypertension and/or systemic hypertension can result in cardiac hypertrophy, cardiac fibrosis and subsequent heart failure. Accumulating results from experimental studies indicate a possible relationship between cardiac hypertrophy and ferroptosis [181,182,183,184,185]. In particular, aortic banding increased cardiac NOX4 expression and reduced cardiac GPX4 activity in animal models indicating that ROS and ferroptosis are likely involved in the process proceeding from hypertrophic cardiomyopathy to heart failure [181,182]. Interestingly, NOX4 knock-down significantly improved left ventricular remodeling and reduced myocytes death through retarded ferroptosis [181]. Similarly, puerarin, a phytoestrogen with antioxidant properties was shown to reduce NOX4 expression and to increase GPX4 expression in the heart failure rat model [182]. In addition, genetic deletion of SLC7A11 in mice was demonstrated to worsen cardiac hypertrophy and fibrosis induced by angiotensin II (Ang II), [183]. Moreover, a very recent study indicates that elabela, a new endogenous ligand for apelin receptor, was able to counteract ferroptosis, myocardial remodeling, fibrosis and heart dysfunction in hypertensive mice [184]. Finally, it has been shown that deletion of NCOA4 in mouse hearts lessened left ventricular chamber dimension and recovered cardiac function together with the reduction in the ferritinophagy-related ferritin breakdown after pressure overload [185]. Moreover, free Fe^2+^ overload and increased ROS generation were inhibited in NCOA4-deficient hearts [185].

#### 3.2.4. Post-Transplant Cardiomyopathy

For patients with severe heart HF, heart transplantation is a significant therapeutic option. Nevertheless, graft damage and graft debacle are remarkable key points capable of causing either short- or long-term unfavorable outcomes in transplanted patients. In the initial phase of heart transplantation, I/R injury is the principal cause of graft dysfunction (GD) in human and animal studies [186]. Subsequently, sterile inflammation is regarded as the main intermediary of GD [187]. In this context, neutrophils are recruited in the injured heart where they induce tissue injury [188]. The current opinion is that neutrophil recruitment during sterile injury is stimulated by the liberation of damage-associated molecular patterns (DAMPs) from dead cells and their binding to receptors of innate immunity, i.e., Toll-like receptors (TLRs) [188,189,190]. Ferroptosis is likely to be the cause of cardiomyocyte death and of DAMP release, which is recognized by TLRs in different heart cells [191]. As for the possible mechanisms causing ferroptosis, Li et al. [188] showed that Fer-1, a well-defined inhibitor of ferroptosis, lowered the levels of hydroperoxy-arachidonoyl-PE and reduced cardiomyocyte cell death and ferroptosis in a heart transplantation mice model indicating that oxidative stress may be the prevailing cause of cardiomyocyte death. It has also been reported that ferroptosis is remarkably linked to endothelial cell dysfunction [91,192] which is characteristically present in GD and is the principal cause of cardiac allograft vasculopathy (CAV), an injury which impacts all parts of the cardiac-vascular tree [193]. Data show that NK cells play a role in CAV formation via a CD8^+^ T-cell mediated mechanism [194] and CD8^+^ T-cells are known to release interferon-gamma which can reduce the synthesis of SLC7A11 and hence favor ferroptosis [195]. Post-transplant cardiomyopathy is a very complex condition with multiple mechanisms involved [138]. Although the role of ferroptosis in post-transplant cardiomyopathy has been established mainly in favoring early inflammation and late immune response, many more aspects should be investigated to have an unambiguous setting.

#### 3.2.5. Iron Overload Cardiomyopathy

While iron is needed for many physiological activities, iron excess may be harmful, as demonstrated in patients with primary (e.g., in patients suffering from hemochromatosis) or secondary iron overload (e.g., in subjects with iron-loading anemias, like β-thalassemia or sickle cell disease receiving regular transfusions) [196]. In these disorders, the heart is one of the major targets for iron accumulation [139], and iron overload cardiomyopathy is the most important cause of morbidity and mortality, being often associated with life-threatening arrhythmias [196,197]. Although iron overload cardiomyopathy has been linked to ferroptosis in animal models [104,198,199,200], the connecting mechanisms between these conditions remain poorly defined. When iron overload is present, the TF-dependent iron uptake is inhibited and NTBI becomes predominant [170], especially in the heart, through the activity of LTCC in HC [201] and of T-type calcium channel (TTCC) in thalassemic cardiomyocytes [202]. LTCC and TTCC activation not only enables extracellular Ca^2+^ entry but also increases the intracellular accumulation of Fe^2+^ through the NTBI uptake [169,170,201,202]. Menon et al. [200] found upregulation of HO-1 in mice with sickle cell disease, a fact that can promote ferroptosis. HO-1 processes heme to biliverdin, carbon monoxide and Fe^2+^, thereby enlarging LIP and promoting ferroptosis [80]. Taken together, these mechanisms are likely to cause an abnormal increase in intracellular LIP. As described above, it is known that mitochondria participate in the process of iron metabolism and that iron overload may be present both at the cellular and mitochondrial levels [78,79,80,81,82,83,84,85]. It has to be mentioned that iron overload is involved in mitochondrial dysfunction and ferroptosis [196,203,204]. Very recently, Xie et al. [205] reported that iron overload boosts ROS generation, depolarizes mitochondrial membrane potential, and dysregulates cytosolic Ca dynamics, a recognized sign of cardiac dysfunction and heart failure [206]. The permeability of mitochondrial transition pores has been indicated as the primary cause of cytosolic Ca dysregulation [207]. In this context, Kumfu et al. [203] have shown that the mitochondrial Ca uniporter (mCU), a protein localized into the inner mitochondrial membrane that carries Ca into mitochondria [208,209], is involved in mitochondrial dysfunction under iron-loaded conditions. Furthermore, Sripetchwandee et al. [210] have shown that mCU plays a major role in cardiac mitochondrial iron uptake under iron-overload conditions. In support of this view, Fefelova et al. [211], by using a mCU knock-out mouse model, demonstrated that iron overload-induced ferroptosis is dependent on the activity of mCU.

In accordance with the influence of mitochondrial LIP in the promotion of ferroptosis, there is also evidence that overexpression of mitochondrial ferritin inhibits the erastin-induced ferroptosis [82]. However, the detailed mechanisms regulating iron metabolism in mitochondria are far from being fully elucidated and further studies are needed to exhaustively understand the precise role of mCU and the crosstalk between Ca and iron. Schematic mechanisms leading to ferroptosis in iron overload cardiomyopathy are shown in Figure 3.

#### 3.2.6. Septic Cardiomyopathy

Septic cardiomyopathy is a severe but potentially reversible complication of sepsis, and one of the major causes of mortality in septic patients [140]. Although cardiomyocyte death is one of the pathogenic hallmarks of septic cardiomyopathy the underlying mechanisms are far from being fully elucidated. Previous reports have shown that lipopolysaccharide (LPS) or stimulator of interferon genes (STING) were directly involved in cardiac dysfunction caused by sepsis through induction of apoptosis autophagy, pyroptosis, or necroptosis [212,213,214,215]. Nevertheless, the data reported that the suppression of regulated cell deaths induced by LPS or STING only partially mitigated the sepsis-induced cardiac injury [188,212,213]. The fact that COX2, a recognized marker of ferroptosis, was found elevated in the heart of a murine model of sepsis [53,216,217] and the evidence that mitochondrial changes induced by LPS were compatible with mitochondrial features of myocardial ferroptosis [29,214], suggest that ferroptosis may be intimately correlated with the development of septic cardiomyopathy caused by LPS or STING. Furthermore, an important contribution to understanding the role and underlying mechanism of ferroptosis on LPS-induced septic cardiomyopathy was provided by Li et al. [218], who demonstrated that LPS was able to promote the expression of NCOA4 in mice. This event reduces the level of ferritin, which was degraded in a ferritinophagy-dependent manner through the interplay between NCOA4 and ferritin, resulting in a higher level of LIP ultimately favoring ferroptosis. In turn, cytoplasmic Fe^2+^ triggers the expression of membrane mitochondrial siderofexin which transfers ferrous ions into mitochondria promoting ROS generation and eventually ferroptosis [218]. Finally, Wang et al. [219], by using a cecal ligation and puncture sepsis cardiac injury mouse model, demonstrated that cecal ligation and puncture increased myocardial marker troponin I level and significantly reduced GPx4 and GSH. Altogether, these results suggest that ferroptosis is one of the main factors leading to septic cardiomyopathy.

#### 3.2.7. Cardiomyopathy Induced by Doxorubicin

Doxorubicin is an effective anthracycline anticancer drug that is still frequently utilized to cure breast cancer, leukemia, and several types of malignancies [220]. However, the clinical utilization of doxorubicin is restricted by its cardiotoxicity, which can lead to irreversible degenerative cardiomyopathy and heart failure [221]. More than 25% of patients taking a cumulative dose of 550 mg/m^2^ of doxorubicin developed heart failure [222]. Fang et al. [10] suggested that ferroptosis pilots doxorubicin-induced cardiomyopathy, on account that cardiomyocytes treated with doxorubicin in mice displayed typical hallmarks of ferroptosis and Fer-1, a ferroptosis inhibitor [223], significantly reduced myocardial injury caused by doxorubicin. The role of iron and ROS in doxorubicin-induced cardiomyopathy development has also been well characterized [224,225,226]. First of all, doxorubicin has been shown to increase LIP [225] and to affect iron homeostasis by inactivating IRP1 and IRP2 activity [226]. Inactive IRPs tie to IREs change the expression of genes implicated in iron metabolism [226]. In particular, doxorubicin disarranges the mRNA linkage of ferritin’s IRE, in this way reducing ferritin H-chain synthesis and favoring the increase in LIP [227]. In addition, doxorubicin increases the expression of TFR1 thus allowing more iron entering into the cells [228]. Similarly, mitochondrial iron overload plays a key role in the mechanisms favoring doxorubicin-induced cardiomyopathy. Heart biopsies from patients with doxorubicin-induced cardiomyopathy-related heart failure showed mitochondria iron overload compared to controls [229], likely due to the doxorubicin-related suppression of ABCB8, a mitochondrial protein that facilitates iron export [229]. Another protein that is crucial in the iron homeostasis of mitochondria is mitochondrial ferritin (MitoFer), which stores ferrous iron [229]. As a matter of fact, the inactivation of MitoFer in rodents favors doxorubicin-induced cardiomyopathy [230]. Free iron binds to doxorubicin giving rise to complexes which create ROS through the Fenton reaction [231]. Doxorubicin, in fact, can bind to Fe^3+^ to create the doxorubicin-Fe3^+^ complex [232,233], which produces the doxorubicin-Fe^2+^complex as a result of either enzymatic or non-enzymatic responses. Doxorubicin-Fe^2+^ complex interacts with oxygen to form O2^•−^ which can be converted into ^•^OH and H_2_O_2_ [234]. H_2_O_2_ can also interact with doxorubicin-Fe^2+^ complexes to produce ^•^OH [235]. Accordingly, secondary to the activity of the doxorubicin-iron complex and the generation of O2^•−^ and ^•^OH, PUFAs are subjected to lipid peroxidation. Moreover, doxorubicin can directly remove Fe^3+^ from ferritin giving rise to doxorubicin-Fe^3+^ complexes, that further grow lipid peroxidation [236]. Finally, doxorubicin can undermine the antioxidant system. A series of studies provide evidence that antioxidant substances such as GPX4, superoxide dismutase, and GSH are significantly reduced in doxorubicin-treated rats and mice [237,238,239], so contributing to lipid peroxidation and ferroptosis. Although the findings described above highlight the importance of ferroptosis in doxorubicin-induced cardiomyopathy, the exact mechanism of doxorubicin-induced cardiomyocyte death is far from being fully elucidated.

#### 3.2.8. Radiation-Induced Cardiomyopathy

Radiation therapy (RT) is frequently employed to cure solid tumors and hematologic malignancies [142]. Radiation-induced heart disease can modify the structure of cardiac tissue and induce pericarditis, coronary artery disease, and congestive heart failure [240]. Radiation-induced cardiomyopathy is a frequent type of radiation-induced heart disease, distinguished by long-term myocardial dysfunction promoted by endothelial damage and myocardial fibrosis [142]. Radiation was initially proven to induce different types of cell death such as apoptosis, necrosis, and autophagic cell death [241,242]. Lately, ferroptosis has also been shown to be linked with radiation-induced cardiomyopathy [243,244,245]. Previous data have demonstrated that vascular injury and endothelial dysfunction play a pivotal role in the development of radiation-induced cardiomyopathy [246,247], due to the basic high sensitivity of endothelial cells to radiation [247]. Mechanistically, it is likely that ROS are the main inducers of endothelial dysfunction and eventually of endothelial cell death caused by radiotherapy [248,249,250]. In this context, it has been reported that radiations can trigger ROS generation by increasing the activity of NOX2 and NOX4 [142,251]. Furthermore, Hu et al. [252] reported that high doses of radiation can damage the mitochondrial membrane of endothelial cells, a fact which can furtherly promote the production of ROS. In addition, it has been reported that ionizing radiation leads to ACSL4 overexpression [243], which is critical for triggering lipid peroxidation and ferroptosis. As mentioned above, ACSL4 has been shown to take part in the generation and assembling of PUFA-PEs in the cell membrane [28], a fact that augments the probability that sensitive PUFAs can be oxidized and eventually triggers ferroptosis [29]. It has been hypothesized that transcriptional factors, such as p53 [48] or chromatin-modifying enzymes such as BRCA1 Associated Protein 1 [253] may be implicated in ionizing radiation-induced ACSL4 overexpression. Finally, RT has also been demonstrated to inhibit SLC7A11 expression both in vitro and in vivo [244]. In this scenario, the mechanisms leading to ferroptosis are paralleled by stimuli resulting in myocardial fibrosis. In particular, the STING pathway, promotes the generation of interferon-gamma [254], and triggers the expression of COX2 [255]. COX2 in turn activates the transforming growth factor-β/hypoxia-inducible factor pathway, a key regulator of fibrosis production, by maximizing the proliferation and differentiation of fibroblasts into myofibroblasts [256].

## 4. Conclusions and Future Directions

CVDs represent the leading cause of death in developed countries. It is well known that the prevention of and reduction in cardiomyocyte death is pivotal to preserve cardiac function and heart failure [6]. In the last decade, accumulating experimental evidence indicate that ferroptosis, a regulated cell death mediated by iron, has a key role in the pathogenesis of myocardial infarction, I/R injury, cardiomyopathies, and heart failure. However, the precise underlying mechanisms in the different CVDs are not completely explained. Moreover, the crosstalk between ferroptosis and other regulated cell death, such as autophagy and apoptosis which has emerged in the last decade [49,69] needs to be furtherly clarified.

Finally, experimental studies clearly demonstrated that targeting ferroptosis could be a new therapeutic option for ischemic heart disease and for many cardiomyopathies. Since iron chelation therapy has been suggested for the treatment of patients with iron overload-related cardiomyopathy, the promising results of preclinical studies open the future possibility to develop effective ferroptosis-specific antagonists also in other clinical settings.

## Figures and Tables

**Figure 1 cells-12-00867-f001:**
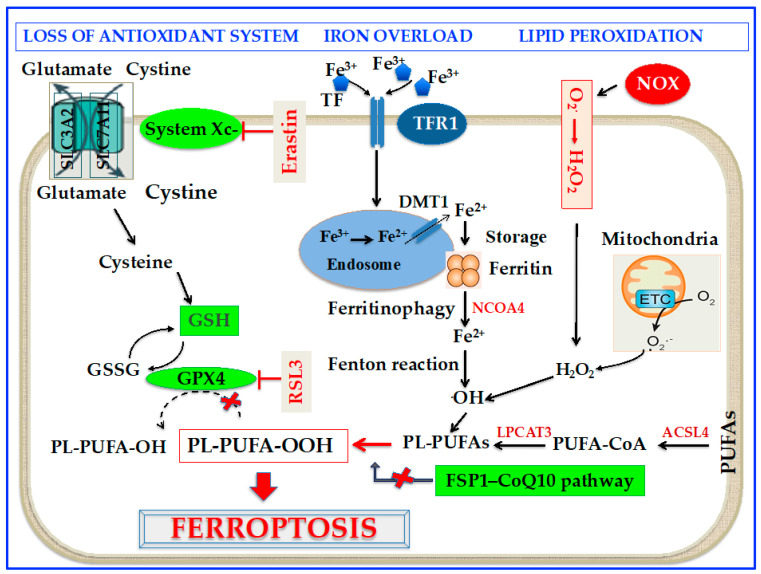
Schematic molecular mechanisms of ferroptosis. *Loss of antioxidant system*: System Xc- facilitates the cellular uptake of cystine, which is reduced by cystine reductase to cysteine, a key precursor in GSH synthesis. GSH serves as an important reducing cofactor for GPX4 (which catalyzes the reduction in lipid peroxides to their corresponding alcohols) while also forming GSSG. Erastin inhibits the cellular uptake of cystine, thus impairing intracellular GSH synthesis. Depletion of GSH leads to the indirect inactivation of GPX4, and the resulting accumulation of lipid peroxides disrupts membrane integrity ultimately leading to ferroptosis. RSL3 acts by directly inactivating GPX4 and does not interfere with the cellular uptake of cystine or intracellular GSH synthesis. *Intracellular iron accumulation*: Circulating iron [Fe^3+^] bound to TF enters into cells by TFR1, localizes in endosomes where it is deoxidized to Fe^2+^. Ultimately, Fe^2+^ is released into a labile iron pool in the cytoplasm by DMT1, while excess iron is stored in ferritin. Under certain circumstances, there may be degradation of ferritin through autophagy, which is termed as “ferritinophagy”, a phenomenon that triggers an increase in labile iron pool and OH via the Fenton reaction. This process requires H_2_O_2_ production by activation of NOX or mitochondria ETC pathways. *Lipid peroxidation*: The generation of PL-PUFAs by ACSL4 and LPCAT3 have a main role in promoting lipid peroxidation. At the final step of ferroptosis, lipid peroxidation directly or indirectly induces pore formation of cell membrane, thus triggering cell death. Abbreviations: ACSL4: Acyl-Coenzyme A synthetase long-chain family member 4; CoQ10: coenzyme Q10 oxidoreductase; DMT1: divalent metal transporter 1; ETC: electron transport chain; FSP1: ferroptosis inhibitor 1; GPX4: glutathione peroxidase 4; GSH: reduced glutathione; GSSG: oxidized glutathione; H_2_O_2_: hydrogen peroxide; LPCAT3: lysophosphatidylcholine acyltransferase 3; NCOA4: nuclear receptor coactivator 4; NOX: nicotinamide adenine dinucleotide phosphate oxidase; O_2_^−^: superoxide; ·OH: hydroxyl radical; PL-PUFA-OH: phospholipid polyunsaturated fatty acid alcohols; PL-PUFA-OOH: phospholipid polyunsaturated fatty acid peroxides; PUFA-CoA: polyunsaturated fatty acid coenzyme A; RSL3: RAS-selective lethal 3; TF: transferrin; TFR1: transferrin receptor 1. *Modified from Fratta Pasini* et al. [22].

**Figure 2 cells-12-00867-f002:**
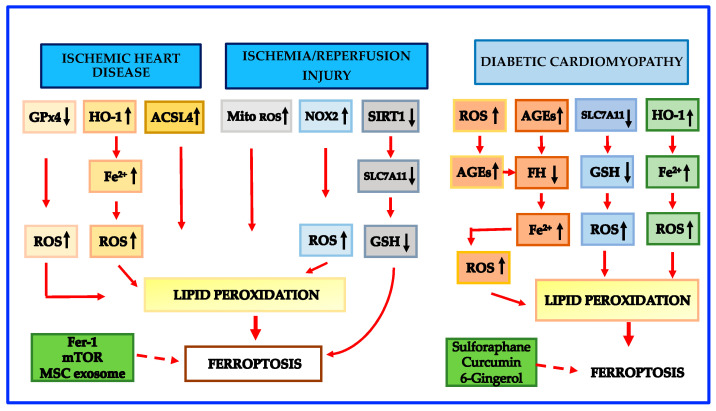
Schematic mechanisms leading to ferroptosis in Ischemic Heart Disease, Ischemia Reperfusion/Injury and Diabetic Cardiomyopathy. Abbreviations: ACSL4: Acyl-Coenzyme A synthetase long-chain family member 4; AGEs: advanced glycation end-products; FER-1: ferrostatin 1; GPX4: glutathione peroxidase 4; GSH: reduced glutathione; HO-1: heme oxygenase-1; MITO ROS: mitochondrial ROS; MSC: mesenchymal stem cell; mTOR: mechanistic target of rapamycin; NCOA4: nuclear receptor coactivator 4; NOX: nicotinamide adenine dinucleotide phosphate oxidase; ROS: reactive oxygen species; SIRT1: sirtuin 1; SLC7A11: solute carrier family 7A member 11 subunit. Legend: dotted arrows = inhibition of ferroptosis.

**Figure 3 cells-12-00867-f003:**
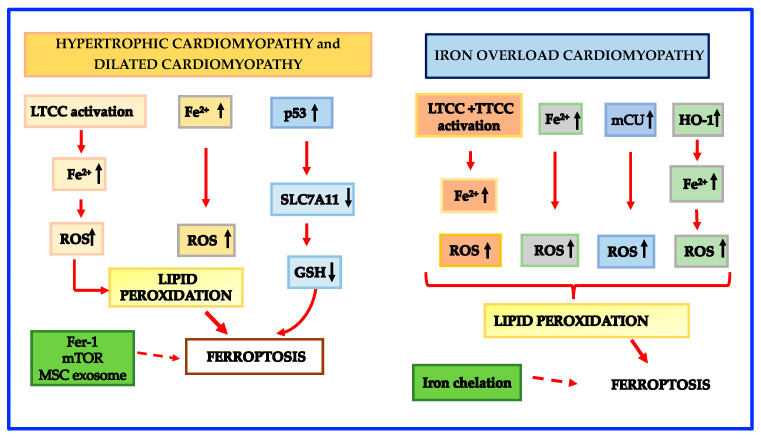
Schematic mechanisms leading to ferroptosis in Hypertrophic, Dilated and Iron overload cardiomyopathies. Abbreviations: GSH: reduced glutathione; HO-1: heme oxygenase-1; LTCC: L-type Ca^2+^ channel; mCU: mitochondrial Ca uniporter; ROS: reactive oxygen species; SLC7A11: solute carrier family 7a member 11 subunit; TTCC: T-type calcium channel. Legend: dotted arrows = inhibition of ferroptosis.

## Data Availability

Not applicable.

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
