# Peer review of "New Insights into the Role of Ferroptosis in Cardiovascular Diseases"

_cells, 2023, doi:10.3390/cells12060867_

Round 1

Reviewer 1 Report

This is a well written article that comprehensively reviews recent progress in molecular mechanisms of ferroptosis and its roles in the death of vascular and cardiac cells. It also covers the pathological relevance of ferroptosis in various cardiovascular disease conditions. Especially the up-to-date research results are provided for diabetic cardiomyopathy, post-transplant cardiomyopathy, septic cardiomyopathy, and radiation-induced cardiomyopathy. This reviewer has only some minor issues.

11) Line 11: consider removing “premature”

22)    Line 37: “Therefore it may have…”  --> therefore ferroptosis may have…

33)    Line 38: “ become”--> becomes

44)    Line 252: is there a full name for “mitoNEET”?

Author Response

  • Line 11: consider removing “premature” 
  • - Line 37: “Therefore it may have…”       --> therefore ferroptosis may have….                                                                

Response:  In the revised manuscript, line 11 and line 37 have been  modified as suggested.

  • Line 38: “ become”--> becomes

Response: We think that in the following sentence “become” is correct : Therefore ferroptosis may have an impact on the mechanisms leading to CVDs and become a suitable target for their prevention.                                                    

  • Line 252: is there a full name for “mitoNEET”?

Response: MitoNEET is the commonly used full name. The mitochondrial membrane protein mitoNEET is  located in the outer mitochondrial membrane. MitoNEET was named according to its C-terminal amino acid sequence, Asn-Glu-Glu-Thr (NEET) (Colca JR, et al. Identification of a novel mitochondrial protein (“mitoNEET”) cross-linked specifically by a thiazolidinedione photoprobe. Am J Physiol Endocrinol Metab. 2004;286:E252–E260

Reviewer 2 Report

This is a well-written and comprehensive review about a very timely topic. The are of course several previous good general reviews about ferroptosis but the focus of the present review on the role of this process in cardiovascular diseases should be of interest to many researchers in the cardiovascular field. The manuscript covers the field well and the illustrations are informative and easy to follow. My only suggestion would be that the authors limit the use of abbreviations. It makes the text difficult to read and some of them appears quite unnecessary. Possible the section describing the role of ferroptosis in atherosclerosis and plaque vulnerability could be somewhat expanded.

Author Response

  • My only suggestion would be that the authors limit the use of abbreviations. It makes the text difficult to read and some of them appears quite unnecessary.

Response: As suggested, we have reduced the number of abbreviations

  • Possible the section describing the role of ferroptosis in atherosclerosis and plaque vulnerability could be somewhat expanded.

Response: We have expanded the section concerning the role of ferroptosis in atherosclerosis and added three new references. In particular: from line 284 to line 285 and from line 288 to line 290 (new ref. 96); from line 307 to line 312 (new refs. 104 e 105).